# Spatio-temporal model to estimate life expectancy and to detect unusual trends at the local authority level in England

Areti Boulieri [ID] , Marta Blangiardo

MRC Centre for Environment and Health, Department of Epidemiology and Biostatistics, School of Public Health, Imperial College London, London, UK

**Correspondence to**
Dr Marta Blangiardo;
m.blangiardo@imperial.ac.uk

## ABSTRACT

**Objectives** To estimate life expectancy at the local authority level and detect those areas that have a substantially low life expectancy after accounting for deprivation.

**Design** We used registration data from the Office for National Statistics on mortality and population in England, by local authority, age group and socioeconomic deprivation decile, for both men and women over the period 2001–2018. We used a statistical model within the Bayesian framework to produce robust mortality rates, which were then transformed to life expectancy estimates. A rule based on exceedance probabilities was used to detect local authorities characterised by a low life expectancy among areas with a similar deprivation level from 2012 onwards.

**Results** We confirmed previous findings showing differences in the life expectancy gap between the most and least deprived areas from 2012 to 2018. We found variations in life expectancy trends across local authorities, and we detected a number of those with a low life expectancy when compared with others of a similar deprivation level.

**Conclusions** There are factors other than deprivation that are responsible for low life expectancy in certain local authorities. Further investigation on the detected areas can help understand better the stalling of life expectancy which was observed from 2012 onwards and plan efficient public health policies.

## INTRODUCTION

The study of life expectancy is of primary interest in public health practice, where it is used to measure the overall health of a population, and also to plan for health and social service provision. Life expectancy has been improving steadily over the last decades, mainly due to changes in behavioural risk factors and healthcare. This phenomenon started to slow down from 2012 onwards and has become cause for concern in several countries, including the USA and the UK.[1 2] Possible explanations of this include strikes in influenza among older people, potentially due to ineffective vaccines,[3] or increases in mortality for mental health conditions, such as cognitive impairment.[4] Recent studies

### Strengths and limitations of this study

► This is the first study of its scale, including more than 8 million records.
► The methodology produces robust estimates via the Bayesian hierarchical framework used.
► Through the detection of local authorities that deviate from the expected trend, our framework helps better understand the stalling of life expectancy in order to implement efficient public health measures.
► In the model, it was not possible to account for population changes due to migration.
► The socioeconomic indicator used was based on data from the last available year; we performed a sensitivity analysis using the midyear deprivation indicator for the period considered.

in the UK found that the excess number of deaths in 2015 was largely driven by the older populations and suggested potential associations with socioeconomic factors, poor health and social care, which can be linked to austerity measures across the health and social care system.[5–7]

Socioeconomic deprivation has been found to be the major determinant of life expectancy by numerous studies over the years, regardless of the measure of deprivation that was used.[8–13] Geographical patterns of life expectancy in England and Wales were found to be mainly attributable to variations in deprivation status in 1998.[8] A more recent study found that the elimination of socioeconomic differences between areas would increase survival among older adults across a number of European countries, with the strongest association in England.[9] The impact of deprivation on overall levels of life expectancy is therefore established. However, the corresponding impact on life expectancy improvements over time is less clear. An early paper found that in most deprived areas improvements in life expectancy were negligible.[10] A more recent paper showed that the life expectancy gap between the most affluent and most deprived areas increased from 2001 to 2016

for both sexes, particularly for women.[11] This finding was not confirmed by later studies where largest improvements in life expectancy were documented in some of the most deprived areas.[14] Recent life expectancy trends by socioeconomic deprivation have mostly been studied by aggregating areas in England into deciles of deprivation at area level.[11 15–17] Limited research has investigated the association between life expectancy and socioeconomic deprivation at the smaller area level, including a study in Scotland[18] and one in Sweden.[19]

It is crucial to focus on small regions and detect those that are most in need. This would help better understand the life expectancy gap and support governments with evidence for public health prioritisation. This cannot be achieved with data description, as splitting the number of deaths by a combination of small area, age, gender and year leads to low number of cases and unstable mortality rates and life expectancy estimates. The need for a modelling approach has been highlighted to overcome this issue.[20] Additionally, we argue that it is important to (1) establish the role of socioeconomic deprivation in detecting areas performing poorly in terms of life expectancy; and (2) detect those areas that differ from others with a similar level of socioeconomic deprivation and show unexplained lower life expectancy. These regions should be investigated further to understand which factors, in addition to deprivation, are driving the stalling of life expectancy.

In this paper, we analysed all-cause mortality counts in England at the local authority level from 2001 to 2018 in order to assess the impact of deprivation on life expectancy improvements over time. Focusing on 2012–2018 when the stalling effect was observed, we identified areas which had lower life expectancy than other areas characterised by a similar level of deprivation.

## METHODS
### Data
We used mortality counts from 2001 to 2018 at the local authority level in England. The local authority level is an administrative subdivision of England with population size ranging from 2300 (Isle of Scilly) to 1.6 million (Kent County Council). In total 315 local authorities were considered, after excluding the Isle of Scilly and the City of London due to their small population. Information on age and sex was available for each record. We considered 20 age groups, each including 5 years except for the first (<1 year old) and the last group (90 years and older), and we analysed men and women separately as these have been previously shown to have very different mortality levels and trends.[21] Population data by age and sex, as well as a measure of deprivation for each local authority were also used for the analysis. All data, originally held by the Office for National Statistics (ONS), were provided to us by Public Health England (PHE).

Deprivation data were available at the decile level. In this study we used the English Index of Multiple Deprivation (IMD), based on seven different domains of deprivation: (1) income deprivation, (2) employment deprivation, (3) education, skills and training deprivation, (4) health deprivation and disability, (5) crime, (6) barriers to housing and services, and (7) living environment deprivation. Each local authority was allocated to a deprivation decile (D1, …, D10) with D1 being the most deprived and D10 the least deprived decile based on the IMD 2019.[22]

### Patient and public involvement
As this was secondary research based on aggregated data that had already been collected, no patients or the public were involved in designing the study, nor were they involved in the analysis or implementation of the study. Dissemination to study participants or patient organisations was not applicable.

### Statistical analysis
We developed a statistical model within a Bayesian hierarchical framework; we specified a Poisson distribution for the mortality counts ($y_{ijtd}$) by age group ($j=1, …, 20$), local authority ($i=1, …, 315$), calendar year ($t=1, …, 18$) and deprivation decile ($d=1, …, 10$):

$$y_{ijtd} \sim \text{Poisson} \left( \lambda_{ijtd} \times \text{Pop}_{ijt} \right)$$

where $Pop_{ijt}$ is the population and $\lambda_{ijtd}$ the mortality rate. Borrowing from the disease mapping literature, the log transformed rate was then modelled as the linear combination of several terms as follows[23 24]:

$$\log \left( \lambda_{ijtd} \right) = a_0 + \alpha_j + \gamma_i + \zeta_t + \phi_d + \omega_{it} + \delta_{ij} + \kappa_{jt} + \nu_{dt}$$

where $\alpha_j$, $\gamma_i$, $\zeta_t$ and $\phi_d$ are random effects identifying the main effects of age, local authority, year and deprivation deciles. Pairwise interactions between age and space ($\delta_{ij}$), age and time ($\kappa_{jt}$), space and time ($\omega_{it}$), and deprivation and time ($\nu_{dt}$) were also included to allow for deviation from the main effects and to increase the level of flexibility of the model.

The Bayesian framework allows the incorporation of assumptions regarding the structure of the mortality rates via prior distributions on the parameters. It is known that mortality rates vary slowly with age, time, and to some extent across space, and it is reasonable to expect that their trends have a smooth pattern. We therefore specified a random walk prior of order 1 (RW1) for the age-specific, time-specific and deprivation-specific components $\alpha_j$, $\zeta_t$, and $\phi_d$, assuming that units that are consecutive in terms of age, time or deprivation decile have similar mortality rates. The spatial component $\gamma_i$ was assigned a conditional autoregressive prior, assuming that neighbouring areas share similar characteristics in terms of mortality rates.[25] We assumed similarity across time and decile of deprivation for each age group, specifying a RW1 on the corresponding interaction terms, $\kappa_{jt}$ and $\nu_{dt}$. On the other hand, the interaction of local authority-age and local authority-year were left unstructured and data driven, through an exchangeable normal prior on $\delta_{ij}$ and

$\omega_{it}$. The overall intercept $a_0$ was assigned a vague normal prior.

A rule to detect local authorities characterised by lower life expectancy was specified based on the space–time interaction term of the model $\omega_{it}$, that, if above 0, would suggest a local increase of mortality rates. We estimated the probability that the space–time interaction term was above 0 for all space–time combinations and we selected the local authorities where this probability exceeded the threshold of 0.95. We restricted the detection on 2012–2018 to focus on the period characterised by the stalling of life expectancy. Further details on the model and the code for running it are available at https://github.com/aretib/LifeExpectancyDetect.

Results are presented as mean estimates and 95% credible intervals. Mortality rates were transformed to life expectancy estimates for each local authority using life tables methods.[21] More details about the procedure to go from mortality rates to life expectancy are reported in https://github.com/aretib/LifeExpectancyDetect.

To evaluate the impact that a change in the definition of the deprivation index has on the detection, we present the results of the same model, where deprivation deciles are based on IMD 2010 (online supplemental material).

## RESULTS

Life expectancy at birth in England was estimated to be 79.6 years for men and 83.2 years for women in 2018. Over the period 2001–2018, life expectancy followed an overall increasing trend, with a flatter pattern from 2012 onwards for both genders, and the gap between women and men decreasing over time. Figure 1 shows the time trends of

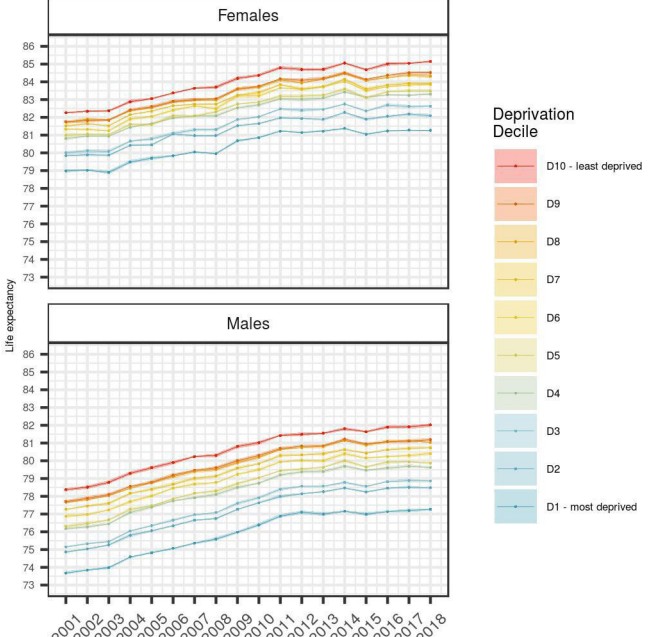

**Figure 1** Life expectancy estimates from 2001 to 2018 across deprivation deciles; the shaded areas correspond to 95% CIs.

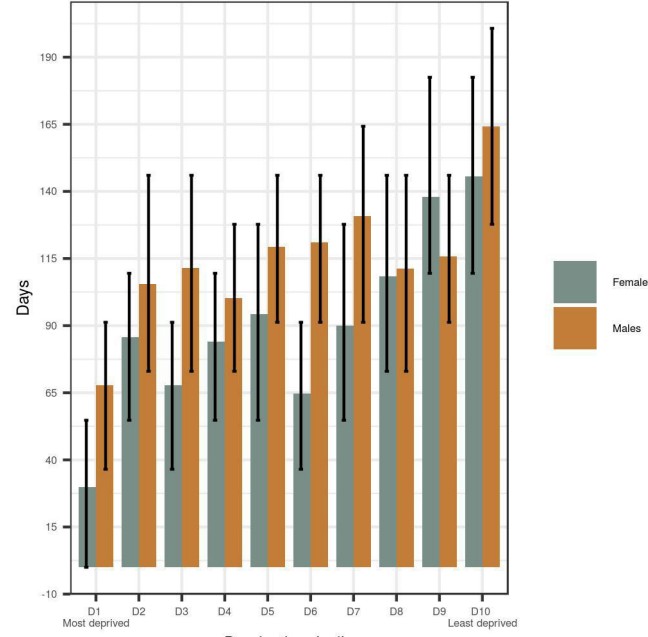

**Figure 2** Life expectancy gap between 2012–2013 and 2017–2018; the error bars correspond to 95% CIs.

life expectancy by deprivation decile group for men and women, respectively. It is observed that the overall level of life expectancy is different across deciles, with higher deprivation corresponding to lower life expectancy. Additionally, it can be noted that life expectancy follows an overall increasing trend for all deciles. However, more deprived groups are characterised by flatter patterns compared with less deprived groups.

Focusing on 2012–2018, figure 2 shows the gap in life expectancy between the beginning and the end of the period and their corresponding 95% interval estimates. There is a difference of more than 100 days in life expectancy gains for the least deprived decile compared with the most deprived decile for both sexes, while it is worth noting the large uncertainty across all the deciles (95% interval estimates depicted by the error bars). When comparing men and women, we observe bigger gains for men across most deprivation deciles. However there is an overlap in the uncertainties for men and women, in line with reports from ONS and PHE.[16 21]

We found variations in life expectancy trends across local authorities, and a number of those were detected as atypical (ie, showing lower life expectancy than in other local authorities in the same deprivation decile) from 2012 to 2018. The maps in figures 3 and 4 show the detected local authorities for men and women, respectively, with the corresponding number of times that these were picked up out of the 7-year period. In particular, 12 unique areas were detected for women and 18 areas for men (tables 1 and 2). Looking at the maps, we do not observe any specific spatial pattern in the detected local authorities, as these are randomly scattered across England. From the tables it can be noted that roughly half of the detected local authorities belong to the two

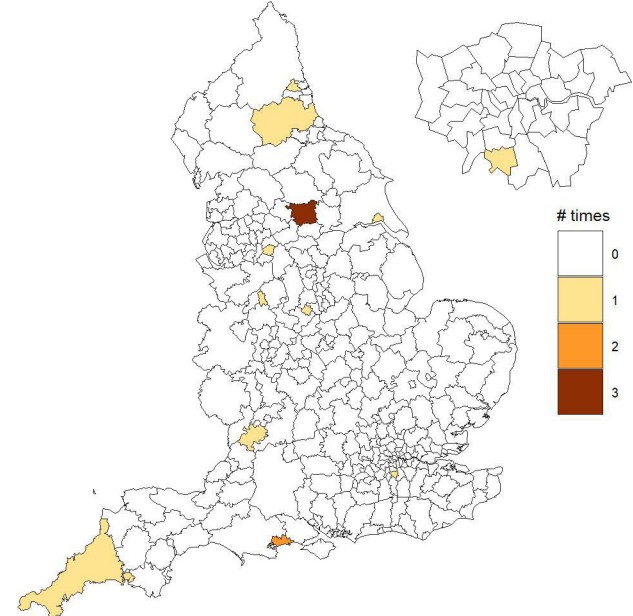

**Figure 3** Local authorities detected with a lower life expectancy than others with a similar deprivation level for women.

most deprived deciles (D1 and D2) for both sexes. Additionally, most local authorities were detected in 1 year only during the 7-year period, with Leeds standing out as it was detected in 2 years for women and 4 years for men. For each detected area, we show mortality rates temporal trends compared with those of the corresponding deprivation decile in online supplemental figures S1–S30.

When running the sensitivity analysis considering the deprivation score from 2010 rather than 2019, we found that the detected areas increase to 31 for women and

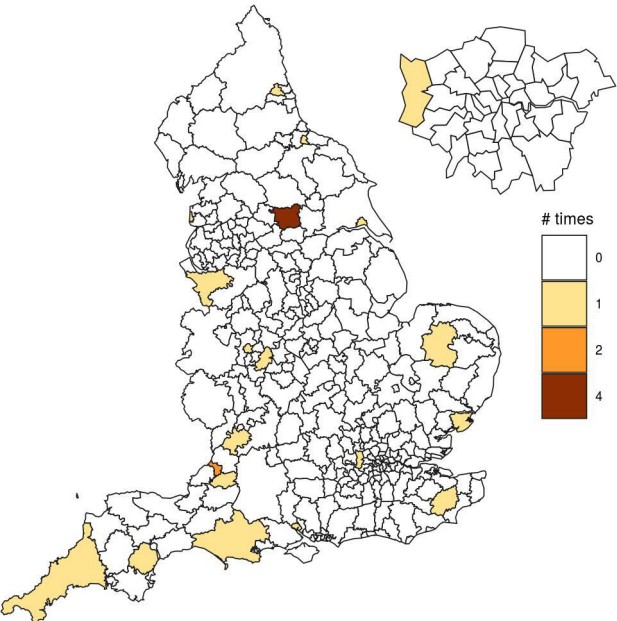

**Figure 4** Local authorities detected with a lower life expectancy than others with a similar deprivation level for men.

**Table 1** Local authorities with a low life expectancy for their deprivation level for women

| Local authority | Decile | Year |
|---|---|---|
| Kingston upon Hull, City of | D1 | 2017 |
| Stoke-on-Trent | D1 | 2018 |
| Tameside | D1 | 2018 |
| Newcastle upon Tyne | D2 | 2015 |
| Plymouth | D2 | 2017 |
| Leeds | D2 | 2015–2018 |
| Derby | D3 | 2018 |
| County Durham | D4 | 2013 |
| Cornwall including Isles of Scilly | D4 | 2016 |
| Bournemouth, Christchurch and Poole | D6 | 2015–2017 |
| Stroud | D9 | 2013 |
| Sutton | D8 | 2014 |

25 for men; for both genders they include all the areas detected in the main model. We present the additional areas detected in online supplemental table S1 and S2, where we include the posterior probability that $\omega_{it}$ is above 0 under the two models. For all the areas, these probabilities are similar and when an area is not detected

**Table 2** Local authorities with a low life expectancy for their deprivation level for men

| Local authority | Decile | Year |
|---|---|---|
| Middlesbrough | D1 | 2017 |
| Wolverhampton | D1 | 2018 |
| Kingston upon Hull, City of | D1 | 2017 |
| Birmingham | D1 | 2014 |
| Tendring | D2 | 2017 |
| Leeds | D2 | 2012, 2015, 2017, 2018 |
| Newcastle upon Tyne | D2 | 2014 |
| Southampton | D2 | 2013 |
| Bristol, City of | D3 | 2013–2018 |
| Cornwall including Isles of Scilly | D4 | 2018 |
| Breckland | D5 | 2018 |
| Ashford | D5 | 2018 |
| Hillingdon | D5 | 2018 |
| Cheshire West and Chester | D6 | 2016 |
| Dorset Council | D7 | 2016 |
| Teignbridge | D7 | 2012 |
| Stroud | D9 | 2016 |
| Bath and North East Somerset | D9 | 2016 |

under the model with IMD 2019, the posterior probability that $\omega_{it}$ is above 0 is not far from the 0.95 threshold, which suggests the results are robust. The 19 additional areas detected for women range across deprivation deciles and 7 move from a less to a more deprived decile between 2010 and 2019; for men 5 additional areas belong to the most deprived deciles and the remaining 2 move from a less to a more deprived deciles between 2010 and 2019.

## DISCUSSION

In this analysis we investigated the life expectancy gap between the most and least deprived local authorities in England. We proposed a statistical tool to look at the life expectancy trends across local authorities and to detect those characterised by lower levels of life expectancy which should be further investigated in order to provide targeted policy measures.

### Principal findings

Our analyses suggest that the increase in life expectancy in England from 2012 to 2018 has been greatest in areas with low levels of deprivation, with more modest increases in life expectancy in areas with high levels of deprivation.[13 21] More specifically, the gain in life expectancy in the wealthiest areas (decile 10) compared with the poorest areas (decile 1) in England was over 100 days for men, and roughly 80 days for women. The differences among deprivation deciles 2–9 were less clear and there was large uncertainty around the estimates.

We found that the life expectancy trends did not follow the same pattern across all local authorities for both men and women between 2012 and 2018. More specifically, we detected the local authorities which depicted a lower life expectancy compared with what would have been expected given their deprivation level. This suggests that factors other than deprivation are potentially responsible for the poor performance of the specific local authorities.

We note here that our study is descriptive as it does not formally assess potential causes in the detected areas. Nevertheless, we expect that it will inform the evidence around life expectancy gaps and the role of deprivation. This should be used as the basis for future studies targeting the detected local authorities to investigate potential responsible factors.

Follow-up research should aim to disentangling the different domains of socioeconomic deprivation, such as barriers to housing and services, or health deprivation and disability. This is needed to better identify if specific domains are responsible for the lower life expectancy observed in the detected local authorities. Methodologically, this requires the use of more complex statistical models, able to account for the potential correlation among the different domains.

Austerity has been linked to the stalling of life expectancy.[26 27] This should be further investigated, particularly looking at how different cities have implemented specific local policies related to austerity which might

have contributed to the different life expectancy trends. These policies are likely to be targeting specific age groups (eg, closure of youth centres or cut to the mental health services or pension credit) and a separate cause-specific analysis may help provide a better picture in this context.

Finally, any events or changes in a neighbourhood should be explored to assess any potential impact these might have on their life expectancy; as an example, life expectancy in an area could be lowered following the opening of a long-term care home, as older people from outside that area move in the facility.

### Comparison to previous studies

Our life expectancy estimates are in line with the ONS estimates for both men and women.[21] In addition, we showed that deprivation has an impact not only on the overall life expectancy estimates but also on life expectancy improvements.[16] This is in line with an earlier study investigating the impact of a health inequalities intervention, implemented by the UK government between 1997 and 2010. The authors found that the life expectancy gap between the population living in the 20% most deprived local authorities and the rest of the population declined during the intervention, while this trend reversed after the intervention ended in 2010.[28]

In contrast to Bennett et al[11] who ran the analyses at the deprivation decile, the novelty of our study lies on the analysis of life expectancy at the local authority level, while accounting for deprivation.[13] We propose a mechanism to detect local authorities in England with lower life expectancy for their deprivation level and therefore provide a first step in the investigation and understanding of factors other than deprivation that might be responsible in order to develop targeted interventions.

### Strengths and limitations

Our analysis of mortality registration data in England from 2001 to 2018 has a number of advantages. First, to our knowledge, it was the first study of its scale, investigating life expectancy at the local authority level in the whole of England over an 18-year period, including more than 8 million records. Second, the methodology that we used is appropriate for the small numbers that arise when mortality counts are considered by local authority, year and age group, and it produces robust estimates, unlike the standard statistical techniques frequently used for this type of analysis.[29] Third, we were able to detect specific local authorities in England which are most left behind in terms of life expectancy improvements, addressing an important gap in the literature. Finally, we proposed a statistical tool that can be used on a routine basis for detection of life expectancy anomalies in England while accounting for deprivation. A segment tool is produced by PHE to provide information on the life expectancy gap at the local authority level.[30] However, this is based on crude mortality rates which are less robust compared with the smoothed ones produced through statistical modelling. Additionally, it provides mean estimates over

2015–2017 rather than annual estimates, and it is not able to automatically flag areas which would require further investigation.

Limitations of our study include population changes because of migration, both within the country and overseas, which were not accounted for in the model. In addition, the allocation of local authorities into discrete groups of deprivation, inevitably yields some variation within local authorities, since not all people can be of the same socioeconomic status, which is a common issue in ecological studies. Lastly, the socioeconomic indicator that we used is based on data from a specific year and does not account for changes in the local authorities between deprivation deciles over the period under study. We chose the deprivation index at 2019 as it provided the most recent estimate of deprivation at small area. Additionally, as the data used to construct the index are related to the period 2013–2017, it is the best representation of deprivation in your detection period (2012–2018). Nevertheless, in our sensitivity analysis we were able to detect all the areas originally detected in the main model and we showed that the results are robust, when comparing the posterior probability that $\omega_{it}$ is above 0.

## CONCLUSIONS

Our study has demonstrated a state-of-the-art statistical tool for detecting specific local authorities in England that have low levels of life expectancy, while accounting for socioeconomic deprivation. There is potential to implement this detection mechanism as part of a routine-monitoring surveillance, in order to focus on specific areas that are most in need.

**Acknowledgements** The authors would like to thank Garyfallos Konstantinoudis, Monica Pirani and Chiara Forlani from Imperial College London, and Allan Baker and Paul Fryers from PHE for their constructive comments which helped improve the paper.

**Contributors** AB and MB contributed to the design of the study, statistical analyses, interpretation and manuscript writing.

**Funding** AB acknowledges support from an MRC Early Career Research Fellowship (MR/M501669/1). MB acknowledges support by Public Health England award titled 'Assessment of Trends in Life Expectancy in England and its Upper-tier Local authorities'. Infrastructure support for the Department of Epidemiology and Biostatistics was provided by the NIHR Imperial Biomedical Research Centre (BRC). This work was part supported by the MRC Centre for Environment and Health, which is currently funded by the Medical Research Council (MR/S019669/1).

**Map disclaimer** The depiction of boundaries on this map does not imply the expression of any opinion whatsoever on the part of BMJ (or any member of its group) concerning the legal status of any country, territory, jurisdiction or area or of its authorities. This map is provided without any warranty of any kind, either express or implied.

**Competing interests** None declared.

**Patient consent for publication** Not required.

**Provenance and peer review** Not commissioned; externally peer reviewed.

**Data availability statement** Data may be obtained from a third party and are not publicly available. The data used for the study were provided by Public Health England (PHE) and cannot be shared online due to confidentiality.

**ORCID iD**
Areti Boulieri http://orcid.org/0000-0003-2022-2236

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
