## [Reviewer comments · BMJ Open]

ARTICLE DETAILS

TITLE (PROVISIONAL)	A spatio-temporal model to estimate life expectancy and to detect unusual trends at the local authority level in England
AUTHORS	Boulieri, Areti; Blangiardo, Marta;

VERSION 1 – REVIEW

REVIEWER	Rosie Seaman University of Stirling, Scotland
REVIEW RETURNED	24-Apr-2020

GENERAL COMMENTS	This is a strong paper. The results are nicely presented and have pragmatic implications. However, it requires further work before publication. The most important issue is the writing. This is both in terms of style and grammar. The introduction is the weakest section of the paper. The main points are there but the writing needs to be improved. The discussion has a number of grammar inconsistencies in the tense used (past or present) when making comparisons with other work. This is not the standard of writing I would expect to see in this level of journal. This paper requires critical reflection and sensitivity analyses surrounding the IMD. What is the justification for using the 2019 IMD as opposed to a midpoint year from the time period covered? Can the authors report the number of LAs which changed deprivation decile - what were the absolute change in LA deciles over the time period? The authors need to reflect on the fact that the IMD health domain contains a measure of mortality and what this means for the results. It would be reassuring to see the authors re-run the analysis using only the income domain of the IMD and if this identifies the same LA. The authors may also want to consider plotting the relationship between the income domain and the overall IMD. Thank you for the opportunity to review such a strong piece of research and I look forward to reading it again.
--

REVIEWER	Stephen Ball Curtin University, Australia
REVIEW RETURNED	26-Apr-2020

GENERAL COMMENTS	Thank you for the opportunity to review this paper. This paper addresses an important topic – in quantifying variation in trends in life expectancy. Overall the paper is well written. However, I do have some major misgivings with the manuscript, which would
---

	need to addressed and re-reviewed, before I would be willing to recommend publication. Comment 1. I found the methods section very opaque, in terms of being able to understand the model structure. At the moment I would think there would be a large amount of guess work required of any reader who wanted to replicate the methods in this paper. I therefore believe the paper would benefit significantly from formally presenting the model structure used (i.e. as one or more equations). In addition to the overall model structure, there are many aspects of the Bayesian modelling that are not presented, but need to be. E.g. the spatial weights used, the estimation process, Bayesian priors used. If there is not enough space in the main paper, or if the authors or journal consider that this breaks the flow of the paper for a broad readership, this should be presented in supplementary material. Until I can really understand the model, I am not in a position to make a reliable review of the rest of the manuscript. Comment 2. I am concerned about the significance of the findings. In the Discussion, under Comparison with Previous Studies, the authors state that “our findings that deprivation has an impact not only on the overall life expectancy estimates but also on life expectancy improvements has been previously shown”, and then state that “the novelty of our study lies on the analysis of life expectancy at the local authority level, while accounting for deprivation”. My concern is that there is not a strong result with the latter. The results indicate 10 local areas that had unusually low life expectancy for females, and 19 local areas with unusually low life expectancy for males. My concern is that we may be seeing a result that is largely due to the equivalent of Type 1 error, i.e. due to chance alone. Given that there are 315 local areas in the dataset, 10 of 315 comprise 3.2%, and 19 of 315 comprise 6.0%. The fact that under or close to 5% of all local areas exceeded the 95% credibility interval makes me wonder if this result can be largely explained by chance alone. In this context, I think it would be worth the authors applying a formal assessment of the significance of the overall variation across the landscape, e.g. by deriving the Deviance Information Criterion to determine the impact of the spatio-temporal term being included. It may be that the main result in terms of geographic variation, is confirmation that deprivation alone does appear to explain the variation. Potentially, this could require changing a lot about who the results and discussion are worded, and how the significance of the paper is presented. Comment 3. Some of the language that the authors use is difficult to follow. E.g. under Principal Findings, the authors state “Our analyses suggest differences in the life expectancy gap between the most and least deprived areas from 2012 to 2018, for both males and females, confirming previous findings”. The use of differences and gap (both forms of difference), in relation to two things (deprivation and time) is hard to follow. Also, this sentence misses important information on the direction of the effect. I wondered if it might help the reader if this was worded as something like the following: “Our analyses suggest that the increase in life expectancy in England from 2012 to 2018 has been greatest in areas with low levels of deprivation, with more modest increases in life expectancy in areas with high levels of deprivation.” Note that this sentence is also used in the abstract.
REVIEWER	Kim WU

REVIEW RETURNED	02-May-2020
-------------

GENERAL COMMENTS	This paper was not possible to account for population changes due to migration in the model. The socio-economic indicator that was used changes in the deprivation level of local authorities over the years was insignificant. Please mention the following article in the discussion. https://bmccgeriatr.biomedcentral.com/articles/10.1186/1471-2318-14-113 https://bmcpublihealth.biomedcentral.com/articles/10.1186/s12889-018-5134-1
---

REVIEWER	Luis Antunes Instituto Português de Oncologia do Porto, Portugal
REVIEW RETURNED	10-May-2020

GENERAL COMMENTS	Where is life expectancy stalling? Detecting local authorities in England across deprivation deciles Comments The authors analyse the trends in life expectancy by local authorities in England during a large period of time. They conclude that life expectancy gains are not homogeneous by deprivation decile. Further they detect local authorities that present lower life expectancies than expected within their deprivation decile. The manuscript is well written and the message is clear. However, I believe the manuscript lacks detail on the type of statistical analysis performed or at least references to it. A reader interested in doing a similar analysis would need more information. Some specific comments to the text can be find below. Abstract:  [ ] It is not clear from the abstract what the period in analysis is. 2001-2018 or 2012-2018?
---

Introduction:

- The sentence “Socio-economic deprivation has been found to be the major determinant of life expectancy by numerous studies over the years, regardless of the measure of deprivation that was used.” is lacking some references.

Methods:

- A description of “local authority level” is missing for readers not familiarised with this concept (size, size variability).
- Why was the IMD for 2019 used? There are different versions of this index across the period 2001-2018. What would be the impact of using other versions of the index?
- The mathematical details of the model used are poorly described and no reference is given that allows the reader to obtain more information.
- Reference [21] does not present details on the method used to calculate life expectancy estimates from mortality rates. Further details on the methodology used are required.

Results:

- The sentence “Focusing on 2012-2018, Figure 2 shows the estimates of life expectancy and their corresponding 95% CI’s” does not describe correctly the content of figure 2. What is represented is the gap in life expectancy between two different periods.
- The description of atypical local authorities would be enriched with life expectancy estimates comparing to the mean of the corresponding deprivation decile. It would give the reader a magnitude of the differences obtained.
- The estimates of life expectancy gaps would be more informative if given with credibility intervals. These are only presented for the point estimates of life expectancy but not for the gaps.

	Discussion:  □ The authors claim to have proposed “a statistical tool that can be used on a routine basis for detection of life expectancy anomalies”. However, no detailed information is provided on the model used neither the software used. It would be very difficult for one to reproduce this analysis with the information provided. Figures:  □ Figure 1: the legend should be “... from 2001 to 2018 ...”.
--	---

VERSION 1 – AUTHOR RESPONSE

Reviewer: 1

Rosie Seaman

This is a strong paper. The results are nicely presented and have pragmatic implications. However, it requires further work before publication.

1) The most important issue is the writing. This is both in terms of style and grammar. The introduction is the weakest section of the paper. The main points are there but the writing needs to be improved. The discussion has a number of grammar inconsistencies in the tense used (past or present) when making comparisons with other work. This is not the standard of writing I would expect to see in this level of journal.

Thank you for pointing this out. We have now reviewed the Introduction and the Discussion and have made some changes, re-writing some sentences in the Introduction and amending the grammar inconsistencies in the Discussion.

2) This paper requires critical reflection and sensitivity analyses surrounding the IMD. What is the justification for using the 2019 IMD as opposed to a midpoint year from the time period covered?

We believe that 2019 is the best one for two reasons: 1) it represents the “latest” level of deprivation, hence it is more relevant in terms of implementation of public health policies; 2) as IMD 2019 is constructed using data collected across 2013-2017 (https://assets.publishing.service.gov.uk/government/uploads/system/uploads/attachment_data/file/833951/loD2019_Technical_Report.pdf), we think it is the best one to represent our detection period (2012-2018).

We have also re-run the analyses using IMD2010 as the mid-point across the entire study period; the model detect all the local authorities which were detected using the main model, together with 19 new areas for females and 7 for males, which we have reported in Supplementary material (Table S1 and S2). We include the posterior probability that is above 0 under the two models. For all the areas, these probabilities are similar and when an area inot detected under the model with IMD 2019, the posterior probability that is above 0 is not far from the 0.95 threshold, which suggests the results are robust. We believe that this sensitivity analysis provided additional evidence that the areas detected using IMD 2019 are unusual and need to be investigated further. We have added a sentence in the

Discussion (lines 287-292) to provide our reasoning for choosing IMD 2019 as the deprivation indicator.

Can the authors report the number of LAs which changed deprivation decile - what were the absolute change in LA deciles over the time period?

We found that 100 areas (33%) change deciles between 2004 and 2019, going down to 60 (20%) between 2015 and 2019. Out of the areas changing deciles, 97 (97%) move one decile up or down between 2004 and 2019, and 58 (99.7%) move one decile up or down between 2015 and 2019. While it is important to acknowledge the fact that our results are based on one year only (which we have done in the Strength and Limitations section of the Discussion), as stressed in the previous point, we believe 2019 is the most representative year for our analysis. Additionally, the fact that the detected areas with IMD2019 are a subset of the ones detected with IMD2010, provides additional evidence on the fact that the detected areas are truly unusual.

The authors need to reflect on the fact that the IMD health domain contains a measure of mortality and what this means for the results. It would be reassuring to see the authors re-run the analysis using only the income domain of the IMD and if this identifies the same LA. The authors may also want to consider plotting the relationship between the income domain and the overall IMD.

Thank you for the interesting comment. We acknowledge that our approach can be used to evaluate the impact of specific domains of deprivation on mortality and consequently life expectancy. Nevertheless, we believe that in the context of the current paper the best approach is to consider the total IMD score, rather than using specific domains, such as income, as this represents only 22.5% of the whole index, and it would discriminate areas only in relation to monetary deprivation. We have produced a scatterplot of IMD2019 vs the income domain and found a correlation of 0.98, suggesting that the results using the income domain would not be substantially different.

Reviewer: 2

Stephen Ball

Thank you for the opportunity to review this paper.

This paper addresses an important topic – in quantifying variation in trends in life expectancy. Overall the paper is well written. However, I do have some major misgivings with the manuscript, which would need to be addressed and re-reviewed, before I would be willing to recommend publication.

1) I found the methods section very opaque, in terms of being able to understand the model structure. At the moment I would think there would be a large amount of guess work required of any reader who wanted to replicate the methods in this paper. I therefore believe the paper would benefit significantly from formally presenting the model structure used (i.e. as one or more equations). In addition to the overall model structure, there are many aspects of the Bayesian modelling that are not presented, but need to be. E.g. the spatial weights used, the estimation process, Bayesian priors used. If there is not enough space in the main paper, or if the authors or journal consider that this breaks the flow of the paper for a broad readership, this should be presented in supplementary material. Until I can really understand the model, I am not in a position to make a reliable review of the rest of the manuscript.

Thank you for this comment. We have now expanded the Methods section (lines 137-161) to include details about the model, including the prior used and some references. We have also provided

a github repository with the code to run the model (lines 168-169).

2) I am concerned about the significance of the findings. In the Discussion, under Comparison with Previous Studies, the authors state that “our findings that deprivation has an impact not only on the overall life expectancy estimates but also on life expectancy improvements has been previously shown”, and then state that “the novelty of our study lies on the analysis of life expectancy at the local authority level, while accounting for deprivation”. My concern is that there is not a strong result with the latter. The results indicate 10 local areas that had unusually low life expectancy for females, and 19 local areas with unusually low life expectancy for males. My concern is that we may be seeing a result that is largely due to the equivalent of Type 1 error, i.e. due to chance alone. Given that there are 315 local areas in the dataset, 10 of 315 comprise 3.2%, and 19 of 315 comprise 6.0%. The fact that under or close to 5% of all local areas exceeded the 95% credibility interval makes me wonder if this result can be largely explained by chance alone.

Thank you for the interesting comment. Whereas we agree that the importance of controlling for multiple testing is clear in classical significance testing, we believe that in predictive setting using hierarchical models the issue is not concerning, as has been shown in the following paper <https://doi.org/10.1080/19345747.2011.618213>. The reason for this is that local predictions from hierarchical models are naturally smoothed towards the global mean, making these consequently less prone to false-positive findings than unsmoothed area-by-area interval estimates.

In this context, I think it would be worth the authors applying a formal assessment of the significance of the overall variation across the landscape, e.g. by deriving the Deviance Information Criterion to determine the impact of the spatio-temporal term being included. It may be that the main result in terms of geographic variation, is confirmation that deprivation alone does appear to explain the variation. Potentially, this could require changing a lot about who the results and discussion are worded, and how the significance of the paper is presented.

We report the DIC in the table below; we found that indeed the inclusion of the space-time interaction leads to a substantial reduction in the DIC for both genders, hence we have kept it in the final model.

Models	Males	Females
Without ST interaction	585551.7	545595.2
With ST interaction	584658.7	544505.1

3) Some of the language that the authors use is difficult to follow. E.g. under Principal Findings, the authors state “Our analyses suggest differences in the life expectancy gap between the most and least deprived areas from 2012 to 2018, for both males and females, confirming previous findings”. The use of differences and gap (both forms of difference), in relation to two things (deprivation and time) is hard to follow. Also, this sentence misses important information on the direction of the effect. I wondered if it might help the reader if this was worded as something like the following: “Our analyses suggest that the increase in life expectancy in England from 2012 to 2018 has been greatest in areas with low levels of deprivation, with more modest increases in life expectancy in areas with high levels of deprivation.” Note that this sentence is also used in the abstract.

Thank you for this, we have changed the sentence as suggested (lines 222-224).

Reviewer: 3

Kim Jong In

1) This paper was not possible to account for population changes due to migration in the model. The

socio-economic indicator that was used changes in the deprivation level of local authorities over the years was insignificant.

In the Limitation section we have acknowledged the fact that we are not able to include migration into the model. We have now re-run the analysis using the IMD2010 as a sensitivity analysis and found consistent results.

2) Please mention the following article in the discussion.

<https://bmcgeriatr.biomedcentral.com/articles/10.1186/1471-2318-14-113>

<https://bmcpublikealth.biomedcentral.com/articles/10.1186/s12889-018-513-1>

We have now included the references in the Introduction (line 81), together with the other papers providing evidence that socio-economic deprivation is a determinant of life expectancy.

Reviewer: 4

Luis Antunes

The authors analyse the trends in life expectancy by local authorities in England during a large period of time. They conclude that life expectancy gains are not homogeneous by deprivation decile. Further they detect local authorities that present lower life expectancies than expected within their deprivation decile.

The manuscript is well written and the message is clear. However, I believe the manuscript lacks detail on the type of statistical analysis performed or at least references to it. A reader interested in doing a similar analysis would need more information.

Some specific comments to the text can be find below.

Thank you for the comment, which was also shared by reviewer 2. We have now re-shaped the Methods (lines 136-161) to include details about the model, including the prior used and some references. We have also provided a github repository with the code to run the model (lines 168-169).

1) Abstract:

It is not clear from the abstract what the period in analysis is. 2001-2018 or 2012-2018?

Thanks for spotting this. The period of analysis is 2001-2018 but the detection period is 2012-2018. We have clarified that in the text (lines 42 and 48).

Introduction:

2) The sentence "Socio-economic deprivation has been found to be the major determinant of life expectancy by numerous studies over the years, regardless of the measure of deprivation that was used." is lacking some references.

We have included several references for this at the end of the sentence, line 81.

3) Methods:

A description of "local authority level" is missing for readers not familiarised with this concept (size, size variability).

We have included it at the beginning of the Data section, page, line 116-118.

Why was the IMD for 2019 used? There are different versions of this index across the period 2001-2018. What would be the impact of using other versions of the index?

Thank you for the very relevant comment, which was shared by reviewer 1. We agree that there is a level of subjectivity in choosing the year for the deprivation index and we acknowledged it in the Limitations section.

We believe that, in the context of the current paper, 2019 is the best one for two reasons: 1) it represents the “latest” level of deprivation, hence it is more relevant in terms of implementation of public health policies; 2) as IMD2019 is constructed using data collected across 2013-2017 (https://assets.publishing.service.gov.uk/government/uploads/system/uploads/attachment_data/file/833951/loD2019_Technical_Report.pdf), we think it is the best one to represent our detection period (2012-2018).

We have also re-run the analyses using IMD2010 as the mid-point across the entire study period; the model detect all the local authorities which were detected using the main model, together with 19 new areas for females and 7 for males, which we have reported in Supplementary material (Table S1 and S2). We include the posterior probability that is above 0 under the two models. For all the areas, these probabilities are similar and when an area is not detected under the model with IMD 2019, the posterior probability that is above 0 is not far from the 0.95 threshold, which suggests the results are robust. We believe that this sensitivity analysis provided additional evidence that the areas detected using IMD 2019 are unusual and need to be investigated further. We have added a sentence in the Discussion (lines 287-292) to provide our reasoning for choosing IMD 2019 as the deprivation indicator.

The mathematical details of the model used are poorly described and no reference is given that allows the reader to obtain more information.

Reference [21] does not present details on the method used to calculate life expectancy estimates from mortality rates. Further details on the methodology used are required.

We have now included more details on the model in the Methods section. The method used to obtain life expectancy from the mortality rates uses life tables and it is the standard used in demography and used in the previous papers (e.g. nb 11 in the reference list). We have now included the code to transform the mortality rates to life expectancy in the github repository (line 172-173).

4) Results:

The sentence “Focusing on 2012-2018, Figure 2 shows the estimates of life expectancy and their corresponding 95% CI’s” does not describe correctly the content of figure 2. What is represented is the gap in life expectancy between two different periods.

Thank you, we have corrected it.

The description of atypical local authorities would be enriched with life expectancy estimates comparing to the mean of the corresponding deprivation decile. It would give the reader a magnitude of the differences obtained.

Thank you for the interesting comment. To provide a richer output on the detected areas we have presented the trends in mortality rates for each area, compared to that of the corresponding deprivation decile. These are in Supplementary material (Figures S1-S30).

The estimates of life expectancy gaps would be more informative if given with credibility intervals. These are only presented for the point estimates of life expectancy but not for the gaps.

The life expectancy gaps in Figure 2 shows the point estimate as well as the 95% credible interval (the black bar).

5) Discussion:

The authors claim to have proposed “a statistical tool that can be used on a routine basis for detection of life expectancy anomalies”. However, no detailed information is provided on the model used neither the software used. It would be very difficult for one to reproduce this analysis with the information provided.

We have now re-shaped the Methods to include more details about the model. We have also added a link to a github repository with the code to run the analysis (lines 168-169).

6) Figures:

Figure 1: the legend should be "... from 2001 to 2018 ...".

Thank you, we have updated it.

VERSION 2 – REVIEW

REVIEWER	Rosie Seaman University of Stirling, Scotland
REVIEW RETURNED	18-Aug-2020

GENERAL COMMENTS	Again, this article is novel and provides important evidence and the amount of time and effort taken to produce this type of analysis should not be underestimated. However, I have ongoing issues with the writing style that mean I cannot yet recommend it for publication. I have provided extensive and detailed comments below section-by-section. Please pay attention to writing in the past tense throughout the manuscript. I do not know if the red sections in the manuscript were intentional, refer to edits made, or is an in error? I prefer to review a clean manuscript. Please pay attention to repetitive language and information. Comments for Strengths and limitations of this study: typos and grammar mistakes throughout "our framework helps to better understand the stalling of life expectancy in order to implement efficient public health interventions." "In the model, it was not possible to account for population changes due to migration." "we performed a sensitivity analysis using the mid-year deprivation indicator" Comments for Introduction: In general, the introduction is weak, has several grammar errors and makes broad claims that I do not think are accurate. Suggested changes: "Life expectancy has been improving steadily over the last decades, mainly due to changes in behavioural risk factors and healthcare. This phenomenon started to slow down from 2012 onwards and has become cause of concern in several country, including the USA and the UK." I would prefer dementia and Alzheimer's disease to be referred to as cognitive impairment. "the excess number of deaths in 2015 was largely driven..." "reductions in spending across the healthcare system" - I do not think it has only been related to the healthcare system. The authors use the phrase social deprivation and area deprivation
--

	interchangeably - this is in correct. Several sentences are too long, unclear and repeat information. General rule if a sentence goes over more than 3 lines of text it should be split into 2 sentences. Suggested changes: "Geographical patterns of life expectancy in England and Wales were found to be mainly attributable to variations in deprivation status in 1998. A more recent study found that the elimination of socio-economic differences between areas would increase survival among older adults across a number of European countries, with the strongest increases estimated for England." "However, the corresponding impact on life expectancy improvements over time is less clear. An early paper found that in most deprived areas improvements in life expectancy were negligible. A more recent paper showed that the life expectancy gap between the most affluent and most deprived areas increased from 2001 to 2016 for both sexes, but the magnitude of the increase was higher for females." Recent life expectancy trends by socio-economic deprivation have mostly been studied by aggregating areas in England into deciles of area deprivation. Limited research has investigated the association between life expectancy and area deprivation at the smaller area level, including a study in Scotland and one in Sweden" "The majority of previous studies have used a low geographical resolution." This sentence is a repetition of the information in the previous sentence and should be deleted. "It is crucial to focus on small region and detect those that are most in need. This would help better understand the life expectancy gap and support governments with evidence for public health policy prioritisation." "This cannot be achieved with data description, as splitting the number of deaths by a combination of small area, age, gender and year leads to low number of cases and unstable mortality rates and life expectancy estimates." "The need for a modelling approach has been highlighted before to overcome this issue. Additionally, we argue that it is important to establish the role of deprivation in detecting areas performing poorly in terms of life expectancy, as well as to detect those areas that differ from others with a similar level of socio-economic deprivation and show unexplained lower life expectancy. These regions should be investigated further to understand which factors, in addition to area deprivation, are driving the stalling of life expectancy." " Focusing on 2012-2018, when the stalling effect was observed, we identified areas which had lower life expectancy than other areas characterised by a similar level of area deprivation but that had higher life expectancy." Comments on Methods: the number of local authorities that were not considered is not discussed or justified.
--	---

	what are the implications of using the full IMD as the health domain includes mortality? "The Bayesian framework allows the incorporation of assumptions regarding the structure of the mortality... rates via prior distributions on the parameters" Comments on Results: "Over the period 2001 to 2018, life expectancy followed an overall increasing trend, with a flatter pattern from 2012 onwards for both genders and the gap between women and men decreasing over time." "Life expectancy follows an overall increasing trend for all deciles. However, more deprived groups are characterised by flatter patterns compared to less deprived groups." "the most deprived one" should be the "the most deprived decile" "When comparing males and females, we observe bigger gains for males across most deprivation deciles. However, there is an overlap in the uncertainties for males and females, in line with reports from ONS and PHE." "(i.e. showing lower life expectancy than in other Local authorities in the same deprivation decile)" - have I understood this correctly? Comments on Discussion: "In this analysis we investigated the life expectancy gap between the most and least deprived local authorities in England. We proposed a statistical tool to look at the life expectancy trends across local authorities and to detect those characterised by lower levels which should be further investigated in order to provide targeted policy interventions." "trends across local authorities and to detect those characterised by lower levels of life expectancy..." Comments on Principal Findings: "...in England was over 100 days more for males, and roughly 80 days more for females." "The differences among deprivation deciles 2 to 9 were less clear and there was large uncertainty around the estimates." "This suggests factors other than deprivation are potentially responsible for the poor performance of the specific local authorities." remove "would call for further investigation by public health bodies and therefore prioritisation for interventions" this is again repetition. " for future studies targeting the detecting local authorities to investigate potential responsible factors." again remove the following information which reads as a hollow
--	---

	recommendation "to be conducted in collaboration with epidemiologists and public health researchers". "The follow up study on the detected areas should consider" - What follow up study are the authors meaning here? suggested changes to this entire paragraph below but overall this paragraph is weak and requires more considered thought. "Follow up research, focusing on the areas we have detected, should aim to disentangling the different domains of socio-economic deprivation, such as barriers to housing and services, or health deprivation and disability. This is needed to better identify which domains are dragging the life expectancy of an otherwise non deprived local authority down." - I do not understand what you mean by "an otherwise non-deprived local authority down" - this is very poorly communicated and needs consideration. "Austerity...." should be the start of a new paragrah. Austerity has been linked to the stalling of life expectancy. This should be further investigated, particularly looking at how different cities have implemented specific local policies related to austerity which might have contributed to the different life expectancy trends. Furthermore, it would worth investigating the different causes of death separately understand if the low life expectancy of an area is attributed to teenage suicides." - This again feels like a hollow recommendation that requires more thought and more well rounded writing. This should be the start of a new paragraph "Finally, any events or changes in a neighbourhood should be explored to assess any potential impact these might have on their life expectancy. An example would be if a hospital that provides long-term care is built in an area, that area may then experience unusually low life expectancy." - I do not necessarily agree can the authors provide a reference detailing how place of death is identified for use with the IMD i.e. is it last known residence of an individual and the deprivation level of that address? Comparison with previous studies "implemented by the UK government between 1997 and 2010. The authors found that ..." "In contrast to Bennett et al. who ran an analyses at the deprivation decile, the novelty of our study lies in the analysis of life expectancy at the local authority level, while accounting for deprivation." Strengths and limitations comments Lots of sentences that should be in the past tense! "Second, the methodology that we used, is appropriate for..." "as it provided the most recent estimate" remove "but one should rather examine these individually."
--	---

	"Third, we were able to" Conclusion comments the first paragraph of the conclusion should be deleted - it is a weak summary and the section paragraph is much more assertive. "Our study has demonstrated a state of the art statistical tool for detecting specific local authorities in England that have low levels of life expectancy, while accounting for socio-economic deprivation. There is potential to use these detection mechanism as part of a routine monitoring surveillance in order to focus on specific areas that are most in need."
--	--

REVIEWER	Jong In Kim Wonkwon University
REVIEW RETURNED	25-Jul-2020

GENERAL COMMENTS	This finding was not confirmed by later studies where the largest improvements in life expectancy were documented in some of the most deprived areas [12]. Please remove this sentence. This research method in this paper is different.
---

REVIEWER	Luis Antunes Portuguese Oncology Institute of Porto, Portugal
REVIEW RETURNED	18-Aug-2020

GENERAL COMMENTS	The authors have replied satisfactorily to the question raised.
---

VERSION 2 – AUTHOR RESPONSE

Reviewer: 1

Rosie Seaman

Again, this article is novel and provides important evidence and the amount of time and effort taken to produce this type of analysis should not be underestimated. However, I have ongoing issues with the writing style that mean I cannot yet recommend it for publication. I have provided extensive and detailed comments below section-by-section. Please pay attention to writing in the past tense throughout the manuscript. I do not know if the red sections in the manuscript were intentional, refer to edits made, or is an in error? I prefer to review a clean manuscript. Please pay attention to repetitive language and information.

We truly appreciate the time the reviewer spent to provide us detailed comments on the style. During the revision process the journal asks to submit a tracked changes version for review (titled BoulieriBlangiardo_tracked, with the edits in red, the one the reviewer refers to) as well as a clean version). We are doing the same this time, hence the reviewer will be able to choose which version she prefers to review.

Comments for Strengths and limitations of this study:

typos and grammar mistakes throughout

"our framework helps to better understand the stalling of life expectancy in order to implement efficient public health interventions."

"In the model, it was not possible to account for population changes due to migration."

"we performed a sensitivity analysis using the mid-year deprivation indicator"

Thank you, we have corrected these.

Comments for Introduction:

In general, the introduction is weak, has several grammar errors and makes broad claims that I do not think are accurate.

Suggested changes:

"Life expectancy has been improving steadily over the last decades, mainly due to changes in behavioural risk factors and healthcare. This phenomenon started to slow down from 2012 onwards and has become cause of concern in several country, including the USA and the UK."

I would prefer dementia and Alzheimer's disease to be referred to as cognitive impairment.

"the excess number of deaths in 2015 was largely driven..."

Thanks, we have changed these as suggested.

"reductions in spending across the healthcare system" - I do not think it has only been related to the healthcare system.

Thanks, we have broadened it to health and social care system.

The authors use the phrase social deprivation and area deprivation interchangeably - this is in correct.

Thanks for this comment. We have changed "area deprivation" to "socio-economic deprivation" throughout the paper.

Several sentences are too long, unclear and repeat information. General rule if a sentence goes over more than 3 lines of text it should be split into 2 sentences. Suggested changes:

"Geographical patterns of life expectancy in England and Wales were found to be mainly attributable to variations in deprivation status in 1998. A more recent study found that the elimination of socio-economic differences between areas would increase survival among older adults across a number of European countries, with the strongest increases estimated for England."

"However, the corresponding impact on life expectancy improvements over time is less clear. An early paper found that in most deprived areas improvements in life expectancy were negligible. A more recent paper showed that the life expectancy gap between the most affluent and most deprived areas increased from 2001 to 2016 for both sexes, but the magnitude of the increase was higher for females."

Recent life expectancy trends by socio-economic deprivation have mostly been studied by aggregating areas in England into deciles of area deprivation. Limited research has investigated the association between life expectancy and area deprivation at the smaller area level, including a study in Scotland and one in Sweden"

"The majority of previous studies have used a low geographical resolution." This sentence is a repetition of the information in the previous sentence and should be deleted.

"It is crucial to focus on small region and detect those that are most in need. This would help better understand the life expectancy gap and support governments with evidence for public health policy prioritisation."

"This cannot be achieved with data description, as splitting the number of deaths by a combination of small area, age, gender and year leads to low number of cases and unstable mortality rates and life expectancy estimates."

"The need for a modelling approach has been highlighted before to overcome this issue. Additionally, we argue that it is important to establish the role of deprivation in detecting areas performing poorly in terms of life expectancy, as well as to detect those areas that differ from others with a similar level of socio-economic deprivation and show unexplained lower life expectancy. These regions should be investigated further to understand which factors, in addition to area deprivation, are driving the stalling of life expectancy."

Thanks, we have changed all these as suggested.

" Focusing on 2012-2018, when the stalling effect was observed, we identified areas which had lower life expectancy than other areas characterised by a similar level of area deprivation but that had higher life expectancy."

We have changed the sentence as suggested, but have removed the last part as we thought it was redundant (line 107-108 BoulieriBlangiardo_main):

" Focusing on 2012-2018, when the stalling effect was observed, we identified areas which had lower life expectancy than other areas characterised by a similar level of deprivation."

Comments on Methods:

the number of local authorities that were not considered is not discussed or justified.

In England there are 317 LAs – as done in most studies, we have excluded the Isle of Scilly and City of London as they have small population size. We have added a sentence on this on line 113-115 (BoulieriBlangiardo_main).

what are the implications of using the full IMD as the health domain includes mortality?

As IMD is widely used as a measure of deprivation in public health research (for instance has been used in the paper by Bennett et al, reference 11 in the paper, and it is considered in the PHE segment tool, reference 30 in the paper), we believe our study should be based on the total IMD score. Using a different score might impact on the classifications of the areas into deciles, hence potentially on the areas detected. A paper by Adams and White (Journal of Public Health 2006 <https://doi.org/10.1093/pubmed/fdl061>), has provided a formal evaluation of the accordance between the full IMD 2004 and the IMD-health domain and found an extremely good agreement between the two. We have done a similar comparison for IMD2019 and found that the correlation across deciles using Kendall τ is 0.85, suggesting a strong agreement, so would not expect substantial changes on the detected areas.

"The Bayesian framework allows the incorporation of assumptions regarding the structure of the mortality..."

Thanks, we have changed as suggested.

Comments on Results:

"Over the period 2001 to 2018, life expectancy followed an overall increasing trend, with a flatter pattern from 2012 onwards for both genders and the gap between women and men decreasing over time."

"Life expectancy follows an overall increasing trend for all deciles. However, more deprived groups are characterised by flatter patterns compared to less deprived groups."

"the most deprived one" should be the "the most deprived decile"

"When comparing males and females, we observe bigger gains for males across most deprivation deciles. However, there is an overlap in the uncertainties for males and females, in line with reports from ONS and PHE."

Thanks, we have changed these as suggested.

"(i.e. showing lower life expectancy than in other Local authorities in the same deprivation decile)" - have I understood this correctly?

Yes the interpretation is correct and we have changed the sentence as suggested.

Comments on Discussion:

"In this analysis we investigated the life expectancy gap between the most and least deprived local authorities in England. We proposed a statistical tool to look at the life expectancy trends across local authorities and to detect those characterised by lower levels which should be further investigated in order to provide targeted policy interventions."

"trends across local authorities and to detect those characterised by lower levels of life expectancy..."

Changed as suggested, thanks.

Comments on Principal Findings:

"...in England was over 100 days more for males, and roughly 80 days more for females."

"The differences among deprivation deciles 2 to 9 were less clear and there was large uncertainty around the estimates."

"This suggests factors other than deprivation are potentially responsible for the poor performance of the specific local authorities." remove "would call for further investigation by public health bodies and therefore prioritisation for interventions" this is again repetition.

We have changed as suggested, thanks.

" for future studies targeting the detecting local authorities to investigate potential responsible factors." again remove the following information which reads as a hollow recommendation "to be conducted in collaboration with epidemiologists and public health researchers".

We do not agree with changing "detected" to "detecting". In this sentence we wanted to stress that the results from our study could be used in the future to investigate factors responsible for the differences in the detected local authorities. We have removed the last part as suggested.

"The follow up study on the detected areas should consider" - What follow up study are the authors meaning here? suggested changes to this entire paragraph below but overall this paragraph is weak and requires more considered thought.

"Follow up research, focusing on the areas we have detected, should aim to disentangling the different domains of socio-economic deprivation, such as barriers to housing and services, or health deprivation and disability. This is needed to better identify which domains are dragging the life expectancy of an otherwise non deprived local authority down." - I do not understand what you mean by "an otherwise non-deprived local authority down" - this is very poorly communicated and needs consideration.

Thanks, we have included the suggested changes; we have also re-shaped the whole paragraph, line 232-236 (BoulieriBlangiardo_main).

"Austerity...." should be the start of a new paragraph.

"Austerity has been linked to the stalling of life expectancy. This should be further investigated, particularly looking at how different cities have implemented specific local policies related to austerity which might have contributed to the different life expectancy trends. Furthermore, it would be worth investigating the different causes of death separately understand if the low life expectancy of an area is attributed to teenage suicides." - This again feels like a hollow recommendation that requires more thought and more well rounded writing.

We have modified the paragraph as follows (lines 237-242 BoulieriBlangiardo_main):

Austerity has been linked to the stalling of life expectancy. This should be further investigated, particularly looking at how different cities have implemented specific local policies related to austerity which might have contributed to the different life expectancy trends. These policies are likely to be targeting specific age groups (e.g. closure of youth centres or cut to the mental health services or Pension credit) and a separate cause-specific analysis may help provide a better picture in this context.

This should be the start of a new paragraph "Finally, any events or changes in a neighbourhood should be explored to assess any potential impact these might have on their life expectancy. An example would be if a hospital that provides long-term care is built in an area, that area may then experience unusually low life expectancy." - I do not necessarily agree can the authors provide a reference detailing how place of death is identified for use with the IMD i.e. is it last known residence of an individual and the deprivation level of that address?

ONS deaths reporting assigns each death to the administrative area based on the last known permanent address of the deceased (report from PHE on this

https://assets.publishing.service.gov.uk/government/uploads/system/uploads/attachment_data/file/831853/Classification_Place_Death_report.pdf

Hence, if a new long-term care home opens in an area, that area might see a drop in life expectancy due to people moving their permanent address to the care home (people moving to the care home might also be previously residents in another administrative area). This is simply due to the care home being located in that area and not related to socio-economic deprivation.

The IMD health domain is also using ONS data, hence the place of death assignment is the same as explained above. A paper by Williams et al., <https://jech.bmj.com/content/58/11/958> reported a negative association between care homes deaths and life expectancy in West Sussex and concluded that similar analyses are needed for larger populations.

Comparison with previous studies

"implemented by the UK government between 1997 and 2010. The authors found that ..."

"In contrast to Bennett et al. who ran an analyses at the deprivation decile, the novelty of our study lies in the analysis of life expectancy at the local authority level, while accounting for deprivation."

Thanks, changed as suggested.

Strengths and limitations comments

Lots of sentences that should be in the past tense!

We have gone through this section to change sentences to past tense if appropriate.

"Second, the methodology that we used, is appropriate for..."

"Third, we were able to"

Changed as suggested, thanks.

"as it provided the most recent estimate"

Changed as suggested, thanks.

remove "but one should rather examine these individually."

Removed as suggested.

Conclusion comments

the first paragraph of the conclusion should be deleted - it is a weak summary and the section paragraph is much more assertive.

Done as suggested.

"Our study has demonstrated a state of the art statistical tool for detecting specific local authorities in England that have low levels of life expectancy, while accounting for socio-economic deprivation. There is potential to use these detection mechanism as part of a routine monitoring surveillance in order to focus on specific areas that are most in need."

Changed as suggested, thanks.

Reviewer: 3

Kim Jong In

This finding was not confirmed by later studies where the largest improvements in life expectancy were documented in some of the most deprived areas [12].

Please remove this sentence. This research method in this paper is different.

We realised there was a typo in the previous version, as the sentence should have included reference 14 rather than 12:

Public Health England. 2019a. "Recent Trends in Mortality in England: Review and Data Packs. A Report on Recent Trends in Life Expectancy and Mortality in England."
<https://www.gov.uk/government/publications/recent-trends-in-mortality-in-england-review-and-data-packs>.

We have corrected it (line 90 BoulieriBlangiardo_main).